# Role of donor nerves in supercharge end-to-side nerve transfer: A rat model study of varying injury severity

Masaru Munemori[1], Akira Kodama [1,2*], Nobuo Adachi[1]

1 Department of Orthopaedic Surgery, Graduate School of Biomedical and Health Sciences, Hiroshima University, Hiroshima, Japan, 2 Division of Regenerative Medicine for Musculoskeletal System, Medical Center for Translational and Clinical Research, Hiroshima University Hospital, Hiroshima, Japan

* akirakodama@hiroshima-u.ac.jp

## Abstract

Supercharge end-to-side (SETS) nerve transfer enhances motor recovery in proximal nerve injuries by providing early reinnervation. However, the optimal indications and mechanisms remain unclear. This study examined the role of donor nerves using rat models of varying injury severity to clarify the clinical indications for SETS. Eighty female Sprague–Dawley rats were assigned to five groups: Control, Mild-SETS(–), Mild-SETS(+), Severe-SETS(–), and Severe-SETS(+). The tibial nerve was transected, decellularized, and reconstructed with a 10 mm (mild) or 20 mm (severe) graft. SETS consisted of end-to-side coaptation of the donor peroneal nerve to the tibial nerve 5 mm distal to the graft. Assessments included the sciatic functional index (SFI; measured every 4 weeks), compound muscle action potentials (CMAPs), gastrocnemius weight, and immunostaining for neurofilament (NF)-positive axons and S100β-positive Schwann cells at 8 and 16 weeks. In mild models, SETS accelerated early recovery in CMAP amplitude and muscle weight without affecting long-term outcomes. In severe models, SETS showed significant increases in CMAP amplitude and muscle weight at 16 weeks. NF-positive axons and S100β-positive Schwann cells increased distal to the coaptation site at 8 and 16 weeks in mild models, whereas both distal and proximal increases were observed in severe models. Donor nerves in SETS enable early arrival of axons and Schwann cells, leading to faster motor improvement. In the long term, spontaneous recovery compensates in mild models, whereas severe models benefit from sustained donor support that promotes regeneration. SETS nerve transfer may therefore be particularly useful in selected mild cases where rapid recovery is desired, and especially in severe cases where spontaneous regeneration is insufficient.

**Data availability statement:** All relevant data are within the manuscript and its Supporting Information file (S1 File).

**Funding:** This study was funded by the Japan Society for the Promotion of Science (JSPS) KAKENHI (Grant Number JP22K09357) to Akira Kodama. The funder had no role in study design, data collection and analysis, decision to publish, or preparation of the manuscript.

**Competing interests:** The authors have declared that no competing interests exist.

## Introduction

Conventional repair techniques for proximal peripheral nerve injuries often result in suboptimal motor recovery due to insufficient trophic support, which leads to degeneration of distal nerve pathways and target muscles before regenerating axons arrive [1–4]. To address this challenge, the supercharge end-to-side (SETS) nerve transfer technique was developed, in which an expendable donor nerve is coapted to the native nerve through surgically created epineurial and perineurial windows [5]. SETS may reduce denervation time in both the distal nerve segment and its target muscles, reproducing the benefits of end-to-end transfer while preserving spontaneous recovery [2,6].

SETS has been applied experimentally in proximal nerve injuries, as an adjunct to nerve repair or grafting, to support recovery when regeneration potential is limited [7,8]. It has also been used clinically for severe cubital tunnel syndrome, with reported improvements in motor function [9,10]. In this condition, even after ulnar nerve decompression at the elbow, regenerated axons must travel long distances, often resulting in irreversible muscular atrophy [3,10]. SETS—from the anterior interosseous nerve to the deep branch of the ulnar nerve—has been proposed to mitigate this by providing early reinnervation. However, its indications remain debated [11]. This uncertainty stems from a lack of investigation into donor nerve effects under varying injury severities, and from incomplete understanding of their impact on native nerve regeneration. While most studies suggest that donor nerves facilitate native nerve regeneration [1,12], some reports raise concerns about interference [13], indicating that the mechanisms are not yet fully understood.

This study aimed to clarify the role of donor nerves in SETS during early and long-term regeneration, using rat models of varying injury severity to identify optimal clinical indications for SETS.

## Materials and methods

### Ethics statement

All procedures were approved by the Hiroshima University Ethics Committee (No. A22-141) and complied with institutional and national guidelines. The study adhered to the ARRIVE guidelines.

### Experimental animals and surgical procedures

A total of 80 female Sprague-Dawley rats (8–12 weeks old, 190–290 g) completed the study. Five additional rats died intraoperatively and were replaced to maintain group sizes, resulting in a total of 85 animals used. The animals were housed in groups of three under standard laboratory conditions. Anesthesia was induced via intraperitoneal injection of xylazine (20 mg/ml; Bayer HealthCare, Leverkusen, Germany) and ketamine (50 mg/ml; Daiichi-Sankyo, Tokyo, Japan). Under sterile conditions, the sciatic nerve was exposed between the sciatic notch and popliteal fossa, followed by neurolysis to isolate the tibial and peroneal nerves over a length of at least 25 mm. The peroneal nerve was transected at the popliteal fossa above the knee in all animals. Except in the control group, the tibial nerve was transected

proximal to the popliteal fossa and in situ decellularization was performed using a previously described freeze-thaw protocol (FTP) [14,15]. The resulting decellularized nerve graft (DNG) was immediately used to bridge the defect and secured with 10−0 nylon sutures. Two neuropathy models of differing severity were established by varying the graft length, as follows: 10 mm for the mild model and 20 mm for the severe model.

## Experimental design and groups

To evaluate the contribution of SETS in both mild and severe models, five experimental groups (n = 8 per group) were established:

1. Control: The tibial nerve remained intact, and the peroneal nerve was transected to eliminate potential donor nerve influence. Both stumps were cauterized, buried in adjacent muscle tissue, and secured with 8−0 nylon sutures.

2. Mild-SETS(−): A 10 mm DNG was inserted without SETS.

3. Mild-SETS(+): A 10 mm DNG was inserted, followed by SETS. Epineural and perineural windows were created in the tibial nerve 5 mm distal to the distal DNG suture site, and the proximal peroneal nerve was coapted with 10−0 nylon.

4. Severe-SETS(−): A 20 mm DNG was inserted without SETS.

5. Severe-SETS(+): A 20 mm DNG was inserted with SETS. (Fig 1)

To standardize the regeneration distances from native and donor nerves to the target muscle, the distal end of the DNG and the SETS coaptation site were aligned across models. In the severe model, the DNG's proximal end was placed near the peroneal bifurcation; in the mild model, it was 10 mm distal to that point. This ensured the distal end—and thus the SETS coaptation site 5 mm beyond it—was anatomically identical in both models.

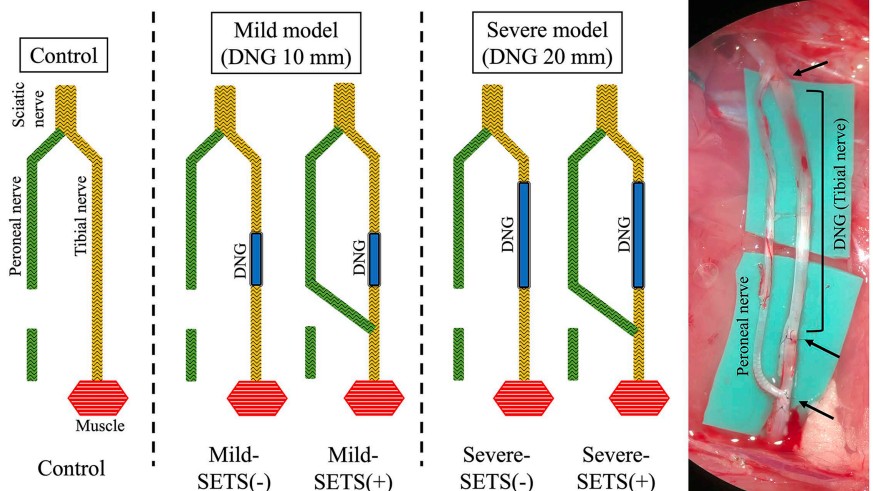

**Fig 1. Experimental design and grouping of SETS models.** Five experimental groups were established: (1) Control: the tibial nerve remained intact, and only the peroneal nerve was transected; (2) Mild-SETS(−): a 10 mm decellularized nerve graft (DNG) was inserted without SETS; (3) Mild-SETS(+): a 10 mm DNG was inserted, followed by SETS; (4) Severe-SETS(−): a 20 mm DNG was inserted without SETS; and (5) Severe-SETS(+): a 20 mm DNG was inserted, followed by SETS. In the SETS procedure, epineurial and perineurial windows were created in the tibial nerve (yellow) approximately 5 mm distal to the distal DNG suture site (blue), and the proximal stump of the peroneal nerve (green) was coapted to the tibial nerve using 10−0 nylon sutures. The schematic diagram (left) and intraoperative photograph (right) illustrate the surgical configuration in the Severe-SETS(+) group. Black arrows indicate the proximal and distal graft anastomoses, as well as the end-to-side coaptation site.

Two time points were selected for evaluation and tissue harvesting: 8 weeks (short-term) and 16 weeks (long-term), each with n = 8 per group, using separate animals for each time point.

All procedures were performed under a surgical microscope using standard microsurgical techniques by a single surgeon.

## Sciatic functional index (SFI)

The sciatic functional index (SFI) was calculated every 4 weeks up to 16 weeks post-implantation to evaluate motor recovery, following the method of Bain et al [16]. Rats were trained to walk along a corridor lined with white paper after dipping their hind paws in black ink to capture footprints. SFI was calculated using the formula:

SFI = −38.3 × (EPL − NPL)/NPL + 109.5 × (ETS − NTS)/NTS + 13.3 × (EIT − NIT)/NIT − 8.8, where EPL, ETS, and EIT are the experimental paw length, toe spread, and intermediary toe spread; NPL, NTS, and NIT are the corresponding values on the contralateral normal side. A score of 0 indicates normal gait; −100 indicates complete functional loss, providing a standardized index of sciatic nerve function.

## Electrophysiological study

Electrophysiological assessments were performed at 8 and 16 weeks postoperatively using validated protocols [17,18]. The sciatic nerve proximal to the neurolysis site was exposed, and needle electrodes were inserted into the gastrocnemius muscle. Bipolar electrodes applied constant current stimulation (2.0 mA, 0.2 ms square-wave pulses) to the sciatic nerve bilaterally. Compound muscle action potentials (CMAPs) were recorded using a Viking Quest system (Nicolet Biomedical, Madison, WI, USA), and onset latency and amplitude were compared bilaterally. Following testing, animals were euthanized with 100% carbon dioxide after intraperitoneal overdose of xylazine and ketamine. To assess the contribution of donor axons, the peroneal nerve was transected at 16 weeks in SETS-treated groups, and recovery rate was defined as the proportion of animals with detectable CMAPs.

## Muscle weight ratio

The gastrocnemius muscles were harvested bilaterally at sacrifice, and the muscle weight ratio was calculated as the ratio of the experimental to contralateral side.

## Immunohistochemistry

Nerves were harvested from three sites surrounding the SETS coaptation site, post-fixed in 4% paraformaldehyde at 4 °C overnight, and immersed in 20% sucrose for 48 h. Each graft was embedded in optimal cutting temperature (OCT) compound (Tissue-Tek, Sakura Finetek, Tokyo, Japan), and frozen sections were prepared using a cryostat microtome (HM520, Thermo Fisher Scientific, Tokyo, Japan). Sections were stained with fluorescent markers and imaged using a BZ-9000 fluorescence microscope (Keyence, Osaka, Japan).

Neurofilament (NF) antibodies labeled axons, and S100β antibodies identified Schwann cells. Sections were fixed in 4% paraformaldehyde/methanol for 30 s, then washed three times with cold PBS for 3 min each. Blocking was performed with 10% normal goat serum (Life Technologies, Carlsbad, CA, USA) at 4 °C for 1 h. Primary antibodies included anti-chicken NF (Abcam, Cambridge, UK, 1:200 in PBS) and anti-rabbit S100β (GeneTex, Irvine, USA, 1:500 in PBS). After overnight incubation at 4 °C, sections were washed three times with PBS and incubated for 1 h with Alexa Fluor 568 anti-chicken IgG and Alexa Fluor 488 goat anti-rabbit IgG secondary antibodies (Thermo Fisher Scientific, 1:200 in PBS). Nuclei were counterstained with DAPI for 10 min at room temperature, then coverslipped.

## Image analysis

In axial sections obtained at 8 and 16 weeks, NF-positive axons and S100β-positive Schwann cells were assessed in three regions around the coaptation site: the proximal tibial nerve (4–5 mm proximal), distal tibial nerve (4–5 mm distal),

and peroneal nerve (4–5 mm proximal). As described previously [17,18], NF-positive axons and Schwann cells were counted at four arbitrarily selected locations within each cross section. Counts per unit area were multiplied by total cross-sectional area to estimate the overall number of NF-positive axons and S100β-positive Schwann cells. Each region measured 800 μm × 600 μm, and images were acquired at 400 × magnification. Quantitative analysis was performed using ImageJ software (NIH, Bethesda, MD, USA).

## Statistical analysis

Data are presented as mean ± standard error. One-way ANOVA followed by Tukey's post hoc test was used for SFI (across all five groups) and for CMAP amplitude, muscle weight ratio, and immunohistochemical outcomes (between SETS(+) and SETS(−) groups within the same severity). For electrophysiological recovery rates, Fisher's exact test was applied. Statistical significance was set at $p < 0.05$. Analyses were performed using EZR version 1.52, a graphical interface for R [19]. The sample size (n = 8 per group) was based on prior studies. Post hoc power analysis using G*Power 3.1 [20,21] showed sufficient power (0.93 and 0.96) for group differences in muscle weight and CMAPs at 16 weeks.

## Results

### Sciatic functional index (SFI)

At 4 and 8 weeks, no significant SFI differences were observed among experimental groups, except the Control group. At 12 weeks, Control and SETS(+) groups were comparable in both models. At 16 weeks, SFI was significantly higher in the Mild-SETS(+) group (−57.9 ± 2.9) than in the Mild-SETS(−) group (−75.8 ± 3.7), and in the Severe-SETS(+) group (−69.6 ± 5.0) than in the Severe-SETS(−) group (−91.4 ± 4.5) (Fig 2).

### Electrophysiological evaluation

Electrophysiological assessments were performed at 8 and 16 weeks. Recovery rate, defined as the proportion of rats with detectable CMAPs, did not differ between the Mild-SETS(+) and Mild-SETS(−) groups at either time point. In contrast, in the severe model, the Severe-SETS(+) group exhibited significantly higher recovery rates than the Severe-SETS(−) group at both 8 ($p < 0.05$) and 16 weeks ($p < 0.01$) (Table 1).

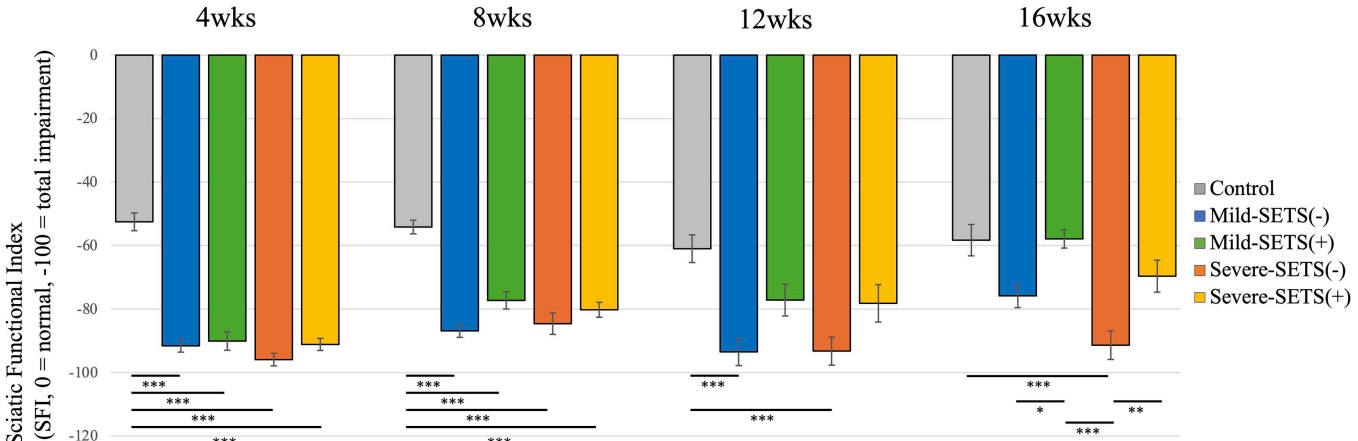

**Fig 2. Functional recovery assessed by the Sciatic Functional Index (SFI).** SFI scores at 4, 8, 12 and 16 weeks postoperatively. The SFI formula is calibrated such that a score of 0 reflects normal sciatic nerve function, whereas −100 corresponds to complete functional loss. *p < 0.05; **p < 0.01; ***p < 0.001 (one-way ANOVA followed by Tukey's post hoc multiple comparisons test).

**Table 1. Recovery rates of CMAP detection at 8 and 16 weeks.**

| Group | | Control | Mild-SETS(-) | Mild-SETS(+) | Severe-SETS(-) | Severe-SETS(+) |
|---|---|---|---|---|---|---|
| Recovery rate (%) | 8Wks | 100 (8/8) | 75.0 (6/8) | 100 (8/8) | 25 (2/8) | 87.5 (7/8)* |
| | 16Wks | 100 (8/8) | 87.5 (7/8) | 100 (8/8) | 25 (2/8) | 100 (8/8)** |
| Recovery rate (%) after donor nerve transection | 16Wks | – | – | 87.5 (7/8) | – | 62.5 (5/8) |

Percentage of rats with detectable CMAPs at 8 and 16 weeks postoperatively (n = 8 per group). *p < 0.05; **p < 0.01 (Fisher's exact test compared with the corresponding non-SETS group).

After donor nerve transection at 16 weeks, recovery rates decreased in both SETS-treated groups, but Severe-SETS(+) still showed a higher rate than Severe-SETS(–) (62.5% vs. 25%).

CMAP latencies did not differ significantly across groups. In the mild model, amplitude at 8 weeks was significantly greater in SETS(+) (p < 0.05), but this difference was not observed at 16 weeks. In contrast, in the severe model at 16 weeks, amplitudes remained significantly higher in SETS(+) than SETS(–) (p < 0.05) (Fig 3).

## Muscle weight ratio

The gastrocnemius muscle weight ratio was expressed as a percentage of the contralateral side. In the mild model at 8 weeks, the ratio was significantly higher in the Mild-SETS(+) group than in the Mild-SETS(–) group (p < 0.05), but this difference disappeared by 16 weeks. In contrast, in the severe model, the Severe-SETS(+) group was significantly higher than Severe-SETS(–) at 16 weeks (p < 0.001) (Fig 4).

## Immunohistochemical evaluation for axonal regeneration

Fig 5 illustrates the three sampling sites and representative immunostaining images at 16 weeks postoperatively. At 8 weeks, distal axon counts in non-SETS groups were 69.7% (3185 ± 677) of controls (4564 ± 217) in the mild model and 49.2% (2245 ± 312) in the severe model. At 16 weeks, recovery in non-SETS groups reached 82.2% (3746 ± 647) in the

CMAP latency ratio (contralateral/ipsilateral)    Peak amplitude ratio (ipsilateral/contralateral)

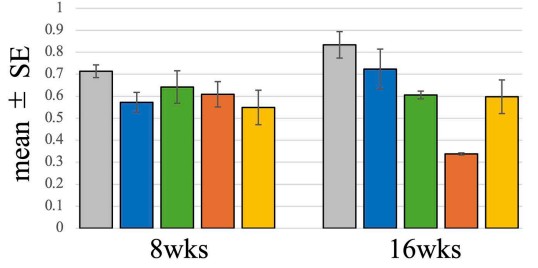 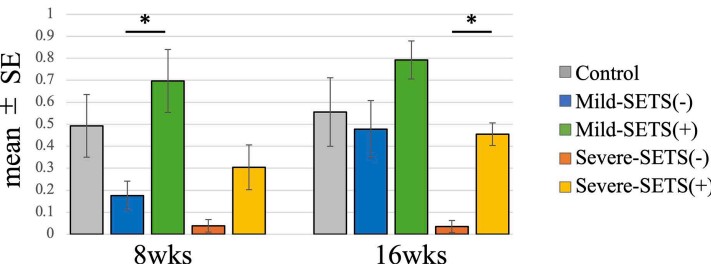

**Fig 3. Electrophysiological evaluation of compound muscle action potentials (CMAPs).** CMAPs of the gastrocnemius muscle were evaluated at 8 and 16 weeks postoperatively. The CMAP latency ratio was calculated as the latency on the experimental side relative to the contralateral side. The peak amplitude ratio was defined as the peak amplitude on the experimental side relative to the maximal amplitude on the contralateral side. Statistically significant differences were determined by one-way ANOVA followed by Tukey's post hoc multiple comparisons test; asterisks indicate only the differences between SETS(–) and SETS(+) groups within the same severity category. *p < 0.05.

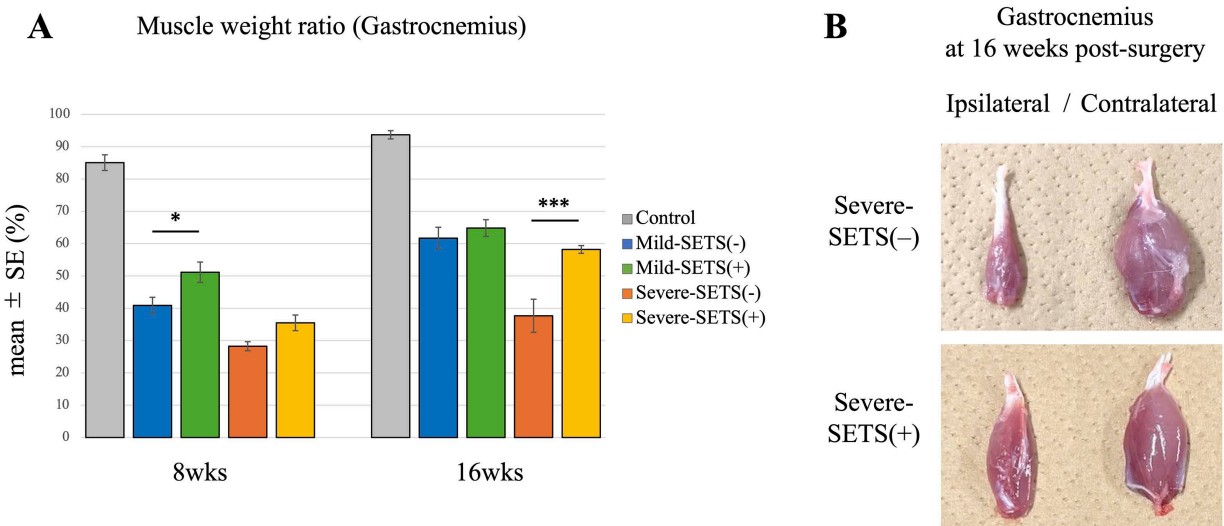

**Fig 4. Evaluation of gastrocnemius muscle recovery.** (A) Gastrocnemius muscle weight ratios at 8 and 16 weeks postoperatively. The ratio was calculated as the weight of the muscle on the operated (ipsilateral) side divided by that on the unoperated (contralateral) side. *p<0.05; ***p<0.001 (one-way ANOVA with Tukey's post hoc test). (B) Representative photographs of gastrocnemius muscles at 16 weeks. Upper panels: Severe-SETS(–) group; lower panels: Severe-SETS(+) group. The operated (ipsilateral) limb is on the left, and the unoperated (contralateral) limb on the right.

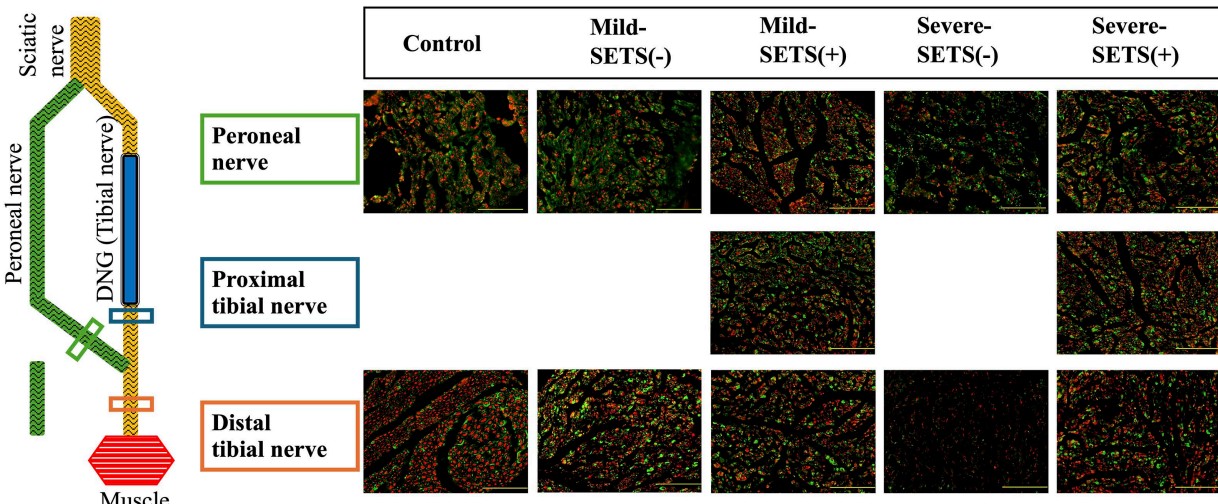

**Fig 5. Immunohistochemical evaluation of axonal regeneration.** Representative immunofluorescence images from three anatomical sites at 16 weeks postoperatively. All sections were obtained in the axial plane and stained for neurofilament (NF, axons; red) and S100β (Schwann cells; green). Images were acquired at 400×magnification. Scale bar=100 μm.

mild model but only 30.4% (1385±262) in the severe model, compared with controls (4555±332) (Table 2). At both 8 and 16 weeks, SETS-treated groups showed significantly higher numbers of NF-positive axons and S100β-positive Schwann cells in the distal tibial nerve than non-SETS groups, regardless of severity (all p<0.05; see Table 2 for full data).

**Table 2. Immunohistochemical evaluation of NF-positive axon counts and S100β-positive Schwann cell counts in three regions surrounding the coaptation site.**

| | | Control | Mild-SETS(-) | Mild-SETS(+) | Severe-SETS(-) | Severe-SETS(+) |
|---|---|---|---|---|---|---|
| **Number of NF-positive axons** | | | | | | |
| **Peroneal nerve** | 8Wks | 1899±339 | 2317±396 | 2065±230 | 2448±254 | 1867±312 |
| | 16Wks | 2410±302 | 2349±471 | 2306±347 | 1854±295 | 1910±232 |
| **Proximal tibial nerve** | 8Wks | - | - | 4918±577 | - | 4168±413** |
| | 16Wks | - | - | 4774±532 | - | 3581±471** |
| **Distal tibial nerve** | 8Wks | 4564±217 | 3185±677 | 5164±333* | 2245±312 | 4957±597** |
| | 16Wks | 4555±332 | 3746±647 | 6959±644*** | 1385±262 | 3962±461** |
| **Number of S100β-positive Schwann cells** | | | | | | |
| **Peroneal nerve** | 8Wks | 1644±245 | 2105±194 | 1831±167 | 2216±367 | 1620±251 |
| | 16Wks | 2009±300 | 2159±421 | 2210±322 | 1782±282 | 1931±183 |
| **Proximal tibial nerve** | 8Wks | - | - | 4241±376 | - | 3683±372** |
| | 16Wks | - | - | 3953±429 | - | 3155±474** |
| **Distal tibial nerve** | 8Wks | 4308±341 | 2676±605 | 4332±273* | 2165±280 | 3793±273* |
| | 16Wks | 4623±267 | 3618±726 | 5466±457* | 1326±192 | 3536±424* |

NF, neurofilament; SE, standard error. Asterisks indicate statistically significant differences (*$p < 0.05$; **$p < 0.01$; ***$p < 0.001$; one-way ANOVA followed by Tukey's post hoc multiple comparisons test). To make a comparison between the SETS(-) and SETS(+) groups, asterisks are shown in the columns of the SETS(+) group. Although all multiple comparisons were performed, only these relevant intergroup differences are displayed in the table.

Moreover, in the severe model, the number of NF-positive axons in the proximal tibial nerve of SETS-treated rats was significantly higher than the distal tibial axon count in non-SETS rats at both time points (8 weeks: 4168±413 vs. 2245±312; 16 weeks: 3581±471 vs. 1385±262; both $p < 0.01$).

## Discussion

This study established severity-defined rat models of incomplete nerve recovery using a simple, reproducible technique. The results demonstrate that the efficacy of SETS depends on injury severity. In mild cases, SETS served as a temporary adjunct that accelerated early recovery without impairing native regeneration. In severe cases, SETS yielded sustained functional and structural benefits. These findings highlight that donor nerves not only provide temporary support but may also facilitate regeneration of the native nerve under conditions of impaired recovery.

Previous models inducing temporary transection or crush injuries [2,5,6,9,11–13,22,23] failed to replicate severity-dependent regeneration due to rats' strong innate recovery capacity [24]. Previous studies have indicated that 20 mm freeze-thawed nerve grafts support approximately 50% axonal regeneration over 12 weeks [15]. Based on this, we employed freeze-thawed decellularized nerve grafts (DNGs) of two lengths (10 mm for mild, 20 mm for severe) as an incomplete regeneration model, serving as a surrogate for chronic compressive neuropathy, which is difficult to reproduce in animals. This approach allowed controlled and reproducible severity-dependent outcomes.

In the mild model, this translated into earlier improvements in compound muscle action potential (CMAP) amplitude, muscle weight ratio and increased distal NF- and S100β-positive cell counts at 8 weeks, consistent with a "babysitting effect" in which donor axons preserve Schwann cell and motor endplate viability until native axons arrive [1,6,12]. Kale et al. [9] also reported that donor-derived axons penetrated the native nerve by day 7 and spread across its cross-section by day 10, with robust distal regeneration by day 35. Similarly, Farber et al. [6] observed increased distal axon counts 5 weeks after SETS. However, these benefits were not sustained, and long-term outcomes were comparable between

SETS-treated and non-SETS groups, suggesting that SETS provides only temporary advantages when spontaneous regeneration is sufficient. In contrast, in the severe model, SETS improved all outcome measures—including axon counts, Schwann cell presence, SFI, CMAP amplitude, muscle weight, and histological regeneration—at 16 weeks. Importantly, even after excluding donor-derived CMAPs, recovery rates remained higher in the Severe-SETS(+) group than in Severe-SETS(−). Similarly, more NF-positive axons were observed in the proximal tibial nerve of SETS-treated rats than in the distal tibial nerve of non-SETS rats, suggesting that donor nerves promoted native axonal regeneration in addition to exerting a "babysitting effect".

In the severe model, the number of NF-positive axons in the distal tibial nerve decreased from 8 to 16 weeks, which may reflect progressive distal degeneration associated with prolonged denervation. Previous studies have shown that chronic denervation impairs Schwann cell support and limits axonal regeneration [25]. In addition, the lower NF counts observed in the peroneal nerve in the severe groups at 16 weeks may reflect altered axonal distribution in response to the regenerative environment, consistent with the concept of preferential reinnervation [26], although this remains speculative. Although some studies raised concerns about donor axons competing for trophic factors or Schwann cells [13], our results—consistent with others [23,27,28]—suggest donor axons can enhance native regeneration under SETS. These discrepancies may stem from injury severity, timing, or technique (e.g., window size/placement). Nadi et al. reported reduced native regeneration in an NIC model, where retrograde tracing at early time points showed approximately 85% donor contribution despite residual native axons [13]. This may reflect limited early extension of native axons rather than true suppression. In contrast, our denervation model largely eliminates residual axons and allows long-term evaluation, demonstrating that donor nerves preserve Schwann cells and facilitate native regeneration [1,27,28].

SETS has been applied in nerve reconstruction, particularly in situations where spontaneous recovery is limited, though its indications remain debated and evidence in mild cases remains limited [1,10,11,29]. Given that anterior interosseous nerve (AIN) harvest at the pronator quadratus level carries minimal morbidity if the pronator teres is preserved [11], SETS may provide early "babysitting" support in selected situations where early recovery is desirable, for example, in cases with progressive muscle atrophy or high functional demands. However, since long-term outcomes were similar between SETS-treated and untreated mild groups, routine use does not appear justified. It should be noted that the present mild model does not directly correspond to clinically defined mild cubital tunnel syndrome, which typically lacks significant axonal degeneration. In contrast, in conditions with impaired regenerative capacity, SETS supported sustained recovery, under-scoring its use as an adjunct to enhance recovery when spontaneous regeneration alone is inadequate. This is consistent with its clinical use in severe cubital tunnel syndrome.

This study has several limitations. First, only female rats were used to minimize variability from hormonal cycles and aggression, but this limits generalizability because sex-based differences in nerve regeneration have been reported [30,31]. Second, the model did not simulate chronic compressive neuropathy, as SETS was applied imme-diately post-injury. While using chronic compressive neuropathy is more clinically relevant, few animal models repro-duce compressive neuropathy with gradable severity [21]. On the other hand, our mild and severe models allow reproducible severity-based evaluation and may be broadly applicable. Third, the short nerve lengths in rodents may allow donor axons to influence native regeneration more easily through local axonal sprouting, a condition that may not directly reflect clinical situations where regeneration occurs over longer distances [24]. Finally, we were unable to distinguish whether distal axons originated from donor or recipient nerves, as retrograde labeling (e.g., Fluoro-Gold tracing) was not performed. Prior studies reported limited retrograde extension of donor axons, with minimal contribu-tion to functional motor units [2,6,9,22]. Thus, the proximal NF-positive axons in our model likely reflect regenerating native axons, but this remains an interpretive limitation. The babysitting effect observed may still hold clinical rele-vance, but whether donor-derived trophic support remains effective over longer distances in clinical settings requires further investigation.

## Conclusions

Donor nerves in SETS provide a "babysitting effect" through early axonal and Schwann cell arrival, leading to earlier motor recovery. In mild injuries, long-term outcomes are achieved by spontaneous recovery; thus, SETS may be considered only in selected cases where rapid recovery is essential. In contrast, its greatest utility lies in severe injuries, where sustained donor support and regenerative effects promote long-term recovery.

## Supporting information

**S1 File. Raw data underlying all quantitative analyses.** Excel file containing individual animal-level data required to reproduce the results. The SFI dataset includes measurements at 4, 8, 12, and 16 weeks, whereas all other outcomes were collected at the terminal time point for each animal (8 or 16 weeks), including electrophysiological parameters (CMAP latency ratio and peak amplitude ratio), muscle weight ratio, and histological measures (NF-positive axons and S100β-positive Schwann cells).
(XLSX)

## Author contributions

**Conceptualization:** Akira Kodama.

**Data curation:** Masaru Munemori.

**Funding acquisition:** Akira Kodama.

**Investigation:** Masaru Munemori.

**Methodology:** Akira Kodama.

**Project administration:** Akira Kodama.

**Resources:** Akira Kodama.

**Supervision:** Akira Kodama, Nobuo Adachi.

**Writing – original draft:** Masaru Munemori.

**Writing – review & editing:** Masaru Munemori, Akira Kodama, Nobuo Adachi.

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
