## [Decision Letter · Decision Letter 0]

15 Feb 2026

Dear Dr. Kodama,

Thank you for submitting your manuscript to PLOS ONE. After careful consideration, we feel that it has merit but does not fully meet PLOS ONE’s publication criteria as it currently stands. Therefore, we invite you to submit a revised version of the manuscript that addresses the points raised during the review process.

We look forward to receiving your revised manuscript.

Kind regards,

Xin Sun, PhD

Staff Editor

PLOS One

Journal Requirements:

https://journals.plos.org/plosone/s/file?id=wjVg/PLOSOne_formatting_sample_main_body.pdf and and and and https://journals.plos.org/plosone/s/file?id=ba62/PLOSOne_formatting_sample_title_authors_affiliations.pdf

3. Please expand the acronym “JSPS” (as indicated in your financial disclosure) so that it states the name of your funders in full.

Reviewers' comments:

Reviewer's Responses to Questions

**Comments to the Author**

1. Is the manuscript technically sound, and do the data support the conclusions?

Reviewer #1: Yes

Reviewer #2: Yes

Reviewer #3: Yes

2. Has the statistical analysis been performed appropriately and rigorously?

Reviewer #1: Yes

Reviewer #2: Yes

Reviewer #3: Yes

3. Have the authors made all data underlying the findings in their manuscript fully available?

Reviewer #1: Yes

Reviewer #2: Yes

Reviewer #3: Yes

4. Is the manuscript presented in an intelligible fashion and written in standard English?

Reviewer #1: Yes

Reviewer #2: Yes

Reviewer #3: Yes

Reviewer #1: Using a rat model with different degrees of nerve injury, the authors showed that donor nerves in supercharge end-to-side (SETS) transfer function differently according to the severity of recipient nerve damage.

Their functional and histological findings indicate that SETS can accelerate early recovery in mild injuries and support long-term regeneration in more severe cases, possibly through a temporary “babysitting” effect on distal targets.

The study is well designed, the data are internally consistent, and the conclusions logically follow from the results.

Reviewer #2: Dear Author,

I had the opportunity to review this study. The constructs and study design are appropriately described and are methodologically sound. The study is well designed, with thoughtfully selected groups and proper implementation of the procedures. The statistical analysis is appropriate and correctly applied.

The Results are presented in a clear and readable manner.

The Discussion section effectively interprets the findings and relates them to the existing literature.

Overall, the manuscript addresses an interesting and relevant research topic and represents a very engaging and valuable contribution to the field.

The manuscript is written in a clear, concise, and focused style and is easy to read. The Introduction adequately introduces the topic, clearly presents the research problem, and outlines the study objectives.

Minor comments:

Figure captions should be placed in a separate section after the References rather than within the main text. This would improve the clarity of the manuscript and help distinguish the main text from figure descriptions.

The figures are of low quality and difficult to read; improvement is required.

Once again, I congratulate you on a well-prepared manuscript.

Reviewer #3: Comments to the authors

Thanks for your effort and contribution in such literature gap

Abstract :

Line no 19- your work is much similar to SETS performed for high ulnar injury and to lesser amount severe entrapment with axonal damage

Introduction :

Paragraph starting from line 48 till 55

You are concentrating on cubital tunnel decompression rather traumatic high ulnar nerve injuries although your study is mimicking traumatic injury and to lesser extent cubital tunnel decompression

It would be much better if u reversed the paragraph

Material and methods :

Line 89

Why you chose to transect the peroneal nerve in the control group ?

Line 146 Immunohistochemistery

How did you obtained nerve tissue from the three sites at 8 weeks without affecting the later on regeneration process

As you stated that the immunohistochemistery photos and count are in the axial plane .

Line 208 :

It should be stated in the methods first that recovery rate will be tested after donor nerve transection rather than to be first stated in the results section

Line 226: muscle weight ratio

How did u obtain the muscle weight in 8 weeks

In the methods u stated that the ratio was done at the sacrifice

So how to be done at 8 weeks then repeated at 16 weeks ?

And how do you explain there is difference in the weight ratio in the control group although the tibial nerve is intact

Table 2 data are confusing .

How do you explain that peroneal nerve NF count is lower in the severe SET - and + groups at 16 weeks time in comparison to control and mild although the difference is in the tibial nerve

Also how the NF count in the severe sets group at distal tibial nerve dropped from 8 weeks (4957) to 3962 at 16 weeks

Discussion

Line 309 in mild cubital tunnel syndrome by definition there is no axonal changes only velocity and latency are affected so in mild entrapment cases there is no justification for any distal SETS .

So your results and recommendation in mild cases could be referred in traumatic cases with more distal sites or small nerve gaps or direct primary repair not in mild entrapment .

Line 320-322

I think it should be reversed

As your study is focusing and mimiking nerve gaps reconstruction and to lesser amount severe cubital tunnel

Line 330 331

What is the transitional relevance ??

.

Reviewer #1: No

Reviewer #2: No

Reviewer #3: **Yes:**Mostafa EzzatMostafa EzzatMostafa EzzatMostafa Ezzat

---

## [Author Response · Author response to Decision Letter 1]

19 Feb 2026

We thank the editor and reviewers for their valuable comments and suggestions.

We have carefully revised the manuscript accordingly.

A detailed point-by-point response is provided in the “Response to Reviewers” document.

We hope that the revised manuscript meets the requirements for publication.

---

## [Decision Letter · Decision Letter 1]

17 Mar 2026

Role of donor nerves in supercharge end-to-side nerve transfer: a rat model study of varying injury severity

PONE-D-25-55769R1

Dear Dr. Kodama,

We’re pleased to inform you that your manuscript has been judged scientifically suitable for publication and will be formally accepted for publication once it meets all outstanding technical requirements.

Kind regards,

Michal Hetman

Academic Editor

PLOS One

Additional Editor Comments (optional):

Reviewers' comments:

Reviewer's Responses to Questions

**Comments to the Author**

Reviewer #1: All comments have been addressed

Reviewer #2: All comments have been addressed

Reviewer #3: All comments have been addressed

2. Is the manuscript technically sound, and do the data support the conclusions?

Reviewer #1: Yes

Reviewer #2: Yes

Reviewer #3: Partly

3. Has the statistical analysis been performed appropriately and rigorously?

Reviewer #1: Yes

Reviewer #2: Yes

Reviewer #3: I Don't Know

4. Have the authors made all data underlying the findings in their manuscript fully available?

Reviewer #1: Yes

Reviewer #2: Yes

Reviewer #3: Yes

5. Is the manuscript presented in an intelligible fashion and written in standard English?

Reviewer #1: Yes

Reviewer #2: Yes

Reviewer #3: Yes

Reviewer #1: The authors appear to have appropriately addressed all comments raised by the reviewers in the previous round, and the revisions seem satisfactory.

Reviewer #2: All suggestions have been implemented and the corrections have been applied. Once again, congratulations on a well-written paper.

Reviewer #3: Dear authors ,

I really appreciate your effort in doing rat model nerve study in a topic i really think is literature gap and would give us valuable data .

And although that i really believe that your final conclusion and findings are right from my point of view regarding SETS transfers

that unclarity and uncertainty in the methodology make that results questionable .

Regarding harvesting samples for immunohistochemistery and muscle weight at different end point 8 weeks and 16 weeks is a major flaw in the methodology, as this means that half number of the rats did not complete the whole study at 16 weeks .

Also if that was the methodology , why that was not stated from the start in the original manuscript although i think it is detail that cannot be forgotten .

I really appreciate your effort but that unclarity in the methodology with confusing results , i am not sure that these results are genuine.

.

Reviewer #1: No

Reviewer #2: No

Reviewer #3: **Yes:**Mostafa ezzatMostafa ezzatMostafa ezzatMostafa ezzat

---

## [Editor Report · Acceptance letter]

PONE-D-25-55769R1

PLOS One

Dear Dr. Kodama,

I'm pleased to inform you that your manuscript has been deemed suitable for publication in PLOS One. Congratulations! Your manuscript is now being handed over to our production team.

Kind regards,

on behalf of

Dr. Michal Hetman

Academic Editor

PLOS One